# The Recretohalophyte *Tamarix TrSOS1* Gene Confers Enhanced Salt Tolerance to Transgenic Hairy Root Composite Cotton Seedlings Exhibiting Virus-Induced Gene Silencing of *GhSOS1*

**DOI:** 10.3390/ijms20122930

**Published:** 2019-06-15

**Authors:** Benning Che, Cong Cheng, Jiajia Fang, Yongmei Liu, Li Jiang, Bingjun Yu

**Affiliations:** 1Lab of Plant Stress Biology, College of Life Sciences, Nanjing Agricultural University, Nanjing 210095, China; 2016116031@njau.edu.cn (B.C.); 2017216030@njau.edu.cn (C.C.); 2017116036@njau.edu.cn (J.F.); 18647979403@163.com (Y.L.); 2Xinjiang Institute of Ecology and Geography, Chinese Academy of Sciences, Urumchi 830011, China; jiangli1015@126.com

**Keywords:** recretohalophyte *Tamarix TrSOS1* gene, salt stress, virus-induced gene silencing (VIGS), hairy root composite cotton plant, yeast mutant, Na^+^ and K^+^ contents

## Abstract

The *salt overly sensitive 1* (*SOS1*) gene encodes the plasma membrane Na^+^/H^+^ antiporter, SOS1, that is mainly responsible for extruding Na^+^ from the cytoplasm and reducing the Na^+^ content in plants under salt stress and is considered a vital determinant in conferring salt tolerance to the plant. However, studies on the salt tolerance function of the *TrSOS1* gene of recretohalophytes, such as *Tamarix*, are limited. In this work, the effects of salt stress on cotton seedlings transformed with tobacco-rattle-virus-based virus-induced gene silencing (VIGS) of the endogenous *GhSOS1* gene, or *Agrobacterium rhizogenes* strain K599-mediated *TrSOS1*-transgenic hairy root composite cotton plants exhibiting VIGS of *GhSOS1* were first investigated. Then, with *Arabidopsis thaliana AtSOS1* as a reference, differences in the complementation effect of *TrSOS1* or *GhSOS1* in a yeast mutant were compared under salt treatment. Results showed that compared to empty-vector-transformed plants, *GhSOS1*-VIGS-transformed cotton plants were more sensitive to salt stress and had reduced growth, insufficient root vigor, and increased Na^+^ content and Na^+^/K^+^ ratio in roots, stems, and leaves. Overexpression of *TrSOS1* enhanced the salt tolerance of hairy root composite cotton seedlings exhibiting *GhSOS1*-VIGS by maintaining higher root vigor and leaf relative water content (RWC), and lower Na^+^ content and Na^+^/K^+^ ratio in roots, stems, and leaves. Transformations of *TrSOS1*, *GhSOS1*, or *AtSOS1* into yeast NHA1 (Na^+^/H^+^ antiporter 1) mutant reduced cellular Na^+^ content and Na^+^/K^+^ ratio, increased K^+^ level under salt stress, and had good growth complementation in saline conditions. In particular, the ability of *TrSOS1* or *GhSOS1* to complement the yeast mutant was better than that of *AtSOS1*. This may indicate that *TrSOS1* is an effective substitute and confers enhanced salt tolerance to transgenic hairy root composite cotton seedlings, and even the *SOS1* gene from salt-tolerant *Tamarix* or cotton may have higher efficiency than salt-sensitive *Arabidopsis* in regulating Na^+^ efflux, maintaining Na^+^ and K^+^ homeostasis, and therefore contributing to stronger salt tolerance.

## 1. Introduction

Soil salinity is one of the major abiotic stress factors that adversely affects plant growth and development, thus reducing crop quality and yield [1,2]. NaCl is the most soluble and abundant salt released into the soil and is the major cause of salt stress on plants or crops [3]. Ionic toxicity, osmotic stress, nutritional imbalance, and oxidative damage are the main causes of salt injury in plants. Under salt stress, plant cells can be protected through several strategies, such as Na^+^ extrusion out of the plasma membrane, ionic imbalances in the vacuole, stress signal transduction, and expression of effector genes encoding ion transporters, channels, enzymes involved in osmolyte biosynthesis and antioxidant systems, etc. and regulatory genes encoding transcription factors, protein kinases, phosphatases, and proteases involved in transcriptional and post-transcriptional regulation as well as in signaling pathways [3,4,5]. In general, ionic toxicity by the accumulation of Na^+^ and Cl^−^ is the primary and dominant factor of salt injury in plants [6], furthermore, crops such as cotton, rice, and barley, are more sensitive to Na^+^ than to Cl^−^ [7]. Many studies have reported on the mechanisms of maintaining low Na^+^ in the cytoplasm from two synergistic aspects, i.e., Na^+^ extrusion out of the cytoplasm into the apoplast and vacuolar compartmentalization through membrane-bound Na^+^/H^+^ antiporters, such as the plasma-membrane-localized salt overly sensitive 1 (SOS1) protein, and the tonoplast-localized NHX1 protein [8].

The *SOS1* gene was originally identified in *Arabidopsis* as one main component of the SOS (salt overly sensitive) signal transduction pathway (also including SOS2 and SOS3) and involved in maintaining cellular Na^+^ homeostasis in plants under salt stress [9,10,11]. SOS1, the activity of which is regulated by the phosphorylation of the SOS2/SOS3 kinase complex, is responsible for extruding Na^+^ from the roots and reducing the Na^+^ content in plants under salt stress. Therefore, *SOS1* is vital for the salt tolerance of plants and has been considered as a superior salt tolerance determinant [12,13]. The physiological roles of the *SOS1* gene have been widely investigated in more than 40 types of plants except *Arabidopsis thaliana*, including glycophytes such as *Oryza sativa* [14], *Solanum lycopersicum* [15], *Chrysanthemum morifolium* [16], *Gossypium hirsutum* [17], *Glycine max*, and *Glycine soja* [18,19] and halophytes such as *Thellungiella salsuginea* [20], *Chenopodium quinoa* [21], *Salicornia brachiata* [22], and *Sesuvium portulacastrum* [23]. Upland cotton (*G. hirsutum*) is one of the most economically important fiber and oil seed crops and can tolerate relatively high salinity and drought stresses compared to other major crops [17,24]. Although the salt tolerance of cotton is moderate and relatively limited, the growth of cotton under salt stress is severely suppressed, especially at the stages of seed germination and young seedlings [25]. The main cause of salinity damage to cotton plants is sodium ion accumulation, which is lethal to most organisms [26]. Chen et al. [17] first cloned the plasma membrane Na^+^/H^+^ antiporter gene *GhSOS1*, and found that *GhSOS1*-transgenic *Arabidopsis* plants displayed enhanced salt tolerance, as indicated by a decreased Na^+^/K^+^ ratio and malondialdehyde (MDA) content in the leaves of the salt-stressed plants. By RNA-Seq analysis of the roots and leaves of diploid cotton species *Gossypium davidsonii* under salt stress and normal conditions, Zhang et al. [27] found lots of differentially expressed genes (DEGs) involved in the maintenance of ion homeostasis and oxidation balance.

Compared to glycophytes, halophytes more efficiently control the influx of Na^+^ into roots, compartmentation of Na^+^ into vacuoles, and distribution of Na^+^ into various tissues. In addition, halophytes have evolved unique anatomical adaptations, such as salt glands and bladders, to excrete Na^+^ out of their body [3,28]. Since halophytes are more capable of overcoming and adapting to soil salinity than glycophytes, studies on the mechanism of the halophytes’ *SOS1* gene in salt tolerance display more academic and practical applications. Oh et al. [20] suggested that, although both *ThSOS1* and *AtSOS1* from the halophyte *T. salsuginea* and its relative species glycophyte *A. thaliana*, respectively, could enhance the salt tolerance of the yeast mutant, the complementary effect of *ThSOS1* was stronger than that of *AtSOS1* under high salinity. Zhou et al. [23] reported that the *SpSOS1* gene from halophyte *S. portulacastrum* complemented the salt sensitivity of *Arabidopsis sos1* mutant plants, and SpSOS1 may have a stronger ability to extrude Na^+^ than the glycophyte *Arabidopsis*. This reveals that the *SOS1* gene of halophytes may play a more critical role in the adaptation to salt stress than that of glycophytes. Therefore, the discovery of salt-tolerance genes from halophytes may have more important theoretical and practical significance that of than those from glycophytes for the in-depth study of the molecular mechanism of plant salt tolerance, especially for the molecular genetic improvement of salt tolerance in crop plants. 

*Tamarix*, a family of woody recretohalophyte species, is widely distributed in the saline soils of drought-stricken areas of Central Asia and China [28]. Our previous studies on three types of *Tamarix* plants (*T. ramosissima*, *T. gansuensis*, and *T. leptostachys*) showed that the growth of *T. ramosissima* was relatively less inhibited, and the expression of *TrSOS1* was upregulated both in roots and shoots under salt stress [29]. However, the unique role of *TrSOS1* in the adaptation of recretohalophyte *T. ramosissima* to high-saline environments remains unclear. *Agrobacterium rhizogenes*-mediated transformation for generating hairy root composite plants has become a powerful tool for studying target gene function and root biology due to the rapidity and simplicity of the method [30,31]. Virus-induced gene silencing (VIGS) is a plant RNA-silencing technique that as a powerful tool for genetic analysis uses viral vectors carrying a fragment of a gene of interest to generate double-stranded RNA, which initiates the silencing of the target gene [32]. In the present work, the effects of *TrSOS1* on the salt sensitivity of *A. rhizogenes* strain K599-mediated *TrSOS1*-transgenic hairy root composite cotton plants, which also exhibited tobacco rattle virus (TRV)-based VIGS of the endogenous *GhSOS1* gene, were first investigated. Then, with *A. thaliana AtSOS1* as a reference, the differences in the complementation effects of the yeast mutant by *TrSOS1* or *GhSOS1* were also analyzed and compared under salt treatment. The objective of this study is to provide a novel theoretical basis for the exploitation of the superior salt tolerance function of the halophyte *SOS1* gene and its future prior utilization in germplasm innovation and genetic improvement of salt tolerant plants or crops by genetic engineering.

## 2. Results

### 2.1. Effects of Salt Stress on the Growth and Related Physiological Parameters of GhSOS1-VIGS Cotton Seedlings

The leaves of cotton seedlings remained green when infected with the empty vector (Vector) for 10 days, while those infected with pTRV2-GhCLA1 (*GhCLA1*-VIGS) displayed a marked leaf-bleached phenotype in the true leaves (Appendix A), in which, GhCLA1 expression was suppressed (Appendix A). This result indicates that the VIGS technique was successfully established in the cotton seedlings and could be further applied to the functional analysis of the target genes *GhSOS1* and *TrSOS1* in this study. Under normal conditions, the cotton seedlings transformed with the empty vector or pTRV2-GhSOS1 (*GhSOS1*-VIGS) also grew well (Appendix A), but GhSOS1 expression was only markedly downregulated in both the true leaves and roots of the latter, thus showing the effective silencing of the *GhSOS1* gene (Appendix A).

When the empty vector and *GhSOS1*-VIGS-transformed cotton plants were exposed to 200 mM NaCl for seven days, the growth of both was obviously inhibited, but the latter displayed smaller leaves, lower plant dry weight and leaf chlorophyll content, and a higher relative electrolytic leakage (REL) value and MDA content in the leaves and roots (Figure 1a–f), which suggests more severe salt injury or sensitivity to the *GhSOS1*-VIGS-transformed cotton plants than the empty-vector-transformed cotton plants. Unlike the changes mentioned above, the root vigor of both cotton plants increased significantly compared with the control, but the increase in the *GhSOS1*-VIGS plants was less than that in the empty-vector-transformed plants (*p* ≤ 0.05). In addition, under NaCl stress, the Na^+^ contents in the roots, stems, and leaves of the empty-vector- and *GhSOS1*-VIGS-transformed cotton plants were substantially increased, and the K^+^ content markedly decreased compared with that of the control (*p* ≤ 0.05). Accordingly, Na^+^/K^+^ ratios in the roots, stems, and leaves were substantially increased. Generally, the variations in the *GhSOS1*-VIGS-transformed plants were substantially larger than those in the empty-vector-transformed plants (Figure 2).

### 2.2. Effects of Salt Stress on the Growth and Related Physiological Parameters of TrSOS1-Transgenic Hairy Root Composite Cotton Plants Exhibiting VIGS of GhSOS1

By PCR amplification and identification of DNA (Appendix A), or RNA (Appendix A) extracted from the roots and leaves of cotton plants, only the expression of *TrSOS1* in the hairy roots and the evident suppression of *GhSOS1* in the roots and leaves indicated that the *TrSOS1*-transgenic hairy root composite cotton plants exhibiting VIGS of *GhSOS1* (named as *hrTrSOS1*-OE/*GhSOS1*-VIGS) were successfully constructed.

Under normal conditions, both the *hrGhSOS1*-VIGS and *hrTrSOS1*-OE/*GhSOS1*-VIGS cotton plants grew well without significant differences in plant dry weight, leaf relative water content (RWC) and chlorophyll content, root vigor, REL value, and MDA content in the roots and leaves (*p* ≥ 0.05) (Figure 3). When exposed to 200 mM NaCl for seven days, the growth of both was clearly reduced and partial true leaves lost water and wilted, but the *hrTrSOS1*-OE/*GhSOS1*-VIGS plants suffered relatively less salt damage (Figure 3a), displaying sufficient substitution and conferring enhanced salt tolerance of *TrSOS1* to the cotton seedlings exhibiting *GhSOS1*-VIGS. In addition, plant dry weight, leaf RWC and chlorophyll content, and root vigor markedly decreased compared with those of the control, whereas the decline in the *hrTrSOS1*-OE/*GhSOS1*-VIGS plants was lower than that of *GhSOS1*-VIGS (Figure 3b–e). The REL value and MDA content in the leaves and roots of both plants observably increased compared with the those of the control, and yet the increase in the *hrTrSOS1*-OE/*GhSOS1*-VIGS plants was also smaller than that of the *hrGhSOS1*-VIGS plants (*p* ≤ 0.05) (Figure 3f,g). The Na^+^ content and Na^+^/K^+^ ratio in the roots, stems, and leaves of both salt-stressed cotton plants greatly increased compared with those of the control, and the K^+^ content in the roots, stems, and leaves visibly decreased. In particular, the increases in the Na^+^ content and Na^+^/K^+^ ratio in the *hrTrSOS1*-OE/*GhSOS1*-VIGS plants were clearly inferior to those of the *hrGhSOS1*-VIGS plants (*p* ≤ 0.05) (Figure 4).

### 2.3. Comparison of TrSOS1, GhSOS1, and AtSOS1 complementation in the yeast NHA1 mutant

To compare the capacity of *TrSOS1*, *GhSOS1*, and *AtSOS1* to substitute for the yeast NHA1 antiporter, mutant complementation tests were performed. All yeast cells, including the wild type G19 (*ena1*), the mutant ANT3 (*ena1nha1*), and the *TrSOS1*-, *GhSOS1*-, or *AtSOS1*-transformed mutants, grew well in YPD (1% yeast extract, 2% peptone, and 2% dextrose), YPG (1% yeast extract, 2% peptone, and 2% galactose), or AP (arginine–phosphate) medium. When exposed to NaCl stress, the growth of the mutant *ena1nha1* was markedly inhibited but was restored by transformation with *TrSOS1*, *GhSOS1*, or *AtSOS1*; however, the recovery of those cells transformed with *TrSOS1* or *GhSOS1* appeared better than that of the cells transformed with *AtSOS1* (Figure 5a). Under NaCl treatment, the cellular Na^+^ content in the mutant *ena1nha1* increased significantly compared with that of the wild type (*ena1*), and the K^+^ content decreased. When *TrSOS1*, *GhSOS1*, or *AtSOS1* was transformed, the increased Na^+^ content and decreased K^+^ level in each transformed yeast mutant were restored to the level of *ena1*; among them, the recovery of the *TrSOS1* or *GhSOS1* transformants was near the control level (*p* ≥ 0.05) and better than that of the *AtSOS1* transformant. Accordingly, the Na^+^/K^+^ ratio in the transformed *ena1nha1* cells substantially decreased to the wild type level though it was still significantly higher than the level in the control (*p* ≤ 0.05), but the recovery of the *TrSOS1* or *GhSOS1* transformants in the context of Na^+^/K^+^ ratio was still superior to that of the *AtSOS1* transformant (*p* ≥ 0.05) (Figure 5b–d).

## 3. Discussion

There have been many studies on the salt tolerance function of the plant *SOS1* gene through *Agrobacterium tumefaciens*-mediated overexpression or RNA interference (RNAi)-based suppression [17,18,20,22,33]. For the first time, a TRV-based VIGS technique for endogenous *GhSOS1* gene silencing, together with the *A. rhizogenes*-mediated hairy root composite plant for target *TrSOS1* gene overexpression, was simultaneously adopted and successfully constructed in this work (Appendix A). When compared to the empty-vector-transformed plants, the *GhSOS1*-VIGS-transformed cotton seedlings were more sensitive to salt stress, and displayed reduced growth, insufficient root vigor, and an increased Na^+^ content and Na^+^/K^+^ ratio in their roots, stems, and leaves (Figure 1 and Figure 2). As a glycophytic plant, cotton normally shows higher salt and drought tolerance than other major crops, and is often considered as a moderately salt-tolerant crop [17,24,28]. Our findings further indicate that, as in the case with the *SOS1* gene in many other plants, the normal expression of *GhSOS1* in the roots and leaves of seedlings is indispensable for the salt tolerance of cotton. Moreover, the overexpression of *TrSOS1* from recretohalophyte *Tamarix* markedly reduced the salt sensitivity and enhanced the salt tolerance of the *GhSOS1*-VIGS cotton plants by maintaining higher root vigor and leaf RWC, and a lower Na^+^ content and Na^+^/K^+^ ratio in the roots, stems, and leaves (Figure 3 and Figure 4). This demonstrates the enhanced complementary function of *TrSOS1*-OE on the salt tolerance of the *GhSOS1*-VIGS cotton plants, and this is also somewhat similar to the complementary effect of the *SOS1* gene from many other glycophytes and a few euhalophytes on the salt sensitivity of the *A. thaliana sos1* mutant [18,23,34].

It is worth noting that the root vigor of both the salt-stressed empty-vector- and *GhSOS1*-VIGS- transformed cotton seedlings increased significantly compared with the control, but the rise of the *GhSOS1*-VIGS plants was less than that of the empty-vector-transformed plants (Figure 1d). However, the root vigor of both the salt-stressed *hrGhSOS1*-VIGS and *hrTrSOS1*-OE/*GhSOS1*-VIGS cotton plants markedly decreased compared with that of the control, whereas the decline in the *hrTrSOS1*-OE/*GhSOS1*-VIGS plants was lower than that of the *hrGhSOS1*-VIGS plants (Figure 3e). The roots of former group of cotton plants were the original roots, which showed a positive salt stress response ability and resulted in enhanced root vigor, while the roots of the latter group were newly grown hairy roots, which showed a weaker salt stress response ability and reduced root vigor. However, further comparison revealed that the root vigor of the *GhSOS1*-VIGS cotton seedlings in former group was lower than that of the empty-vector-transformed seedlings under salt stress, indicating a positive correlation between the lower root vigor of cotton plants and VIGS-mediated endogenous *GhSOS1* suppression. In contrast, the root vigor of the *hrTrSOS1*-OE/*GhSOS1*-VIGS cotton plants in latter group was higher than that of the *hrGhSOS1*-VIGS plants under salt stress, also indicating a positive correlation between higher root vigor of cotton plants and exogenous *TrSOS1* overexpression. Generally, the upregulation of the expression level of a functional gene, such as *SOS1*, is conducive to the activity enhancement of its encoded protein [35]. The expression level of the *TrSOS1* gene in the roots and/or leaves of cotton plants, and even the transport activity of the SOS1 protein seem to be positively correlated with root vigor to a certain extent. However, the details remain to be further studied.

Under saline conditions, salt excretion was evidently observed on the surface of the aboveground part of three species of recretohalophyte *Tamarix* [29]. Peng et al. [25] reported that glandular trichomes (GTs) on the leaves of salt-treated cotton plants could secrete excess salt. With *A. thaliana AtSOS1* as a reference, we found that the transformation of *TrSOS1, GhSOS1* or *AtSOS1* into the yeast mutant ANT3 (*ena1nha1*) reduced the cellular Na^+^ content and Na^+^/ K^+^ ratio, increased the K^+^ level under salt stress, and displayed good growth complementation effect in saline conditions. In particular, the complementation ability of *TrSOS1* or *GhSOS1* in the salt tolerance of the yeast mutant was evidently superior to that of *AtSOS1*, but the ability of *TrSOS1* was basically equal to or slightly better than that of *GhSOS1* (Figure 5). This suggests that the contribution of the *SOS1* genes from the recretohalophyte *Tamarix*, salt-tolerant glycophyte cotton, and salt-sensitive glycophyte *Arabidopsis* to salt tolerance is *TrSOS1* ≥ *GhSOS1* > *AtSOS1* by the yeast mutant complementation test, which may be another innovative discovery in this study. Oh et al. [20] reported that although both *ThSOS1* and *AtSOS1* from the halophyte *T. salsuginea* and glycophyte *A. thaliana*, respectively, could enhance the salt tolerance of the yeast mutant, the complementary effect of *ThSOS1* was stronger than that of *AtSOS1* under high salinity. Therefore, this study can indicate that the salt-tolerant function of the *SOS1* gene of a certain kind of plant, may be related to its habitat or salt-tolerant habit, and the molecular evolutions of the *SOS1* gene of halophytes, salt-tolerant and salt-sensitive glycophytes need further attention and study.

## 4. Materials and Methods

### 4.1. Plant Materials, Bacteria and Yeast Strains, and Plasmids

Plants including *T. ramosissima*, *G. hirsutum* cv. Xinluzao No. 51, and *A. thaliana* wild type (WT) (Columbia ecotype glabrous 1), *Escherichia coli* DH5α, *A. tumefaciens* strain GV3101, *A. rhizogenes* strain K599, *Saccharomyces cerevisiae* strain G19 (*Δena1::HIS3::ena4*, represented as *ena1*), and its plasma membrane antiporter NHA1 mutant ANT3 (*Δena1::HIS3::ena4, Δnha1::LEU2*, represented as *ena1nha1*), the plant transformation binary vector pSuper1300^+^, the pTRV1 and pTRV2 vectors and yeast expression vector plasmid pYES2 were used in this study.

### 4.2. TrSOS1, GhSOS1, and AtSOS1 Gene Cloning and Vector Construction

Seeds of *T. ramosissima* and *G. hirsutum* cv. Xinluzao No. 51 were sterilized and sown in pots containing a sterilized peat moss and vermiculite mixture and grown in a growth chamber under a 12 h light/12 h dark cycle at 28 ± 2 °C, with 60–70% relative humidity. In addition, seeds of *A. thaliana* WT were first chilled at 4 °C in the dark for two days, and sown and grown in a growth chamber under a 14 h light/10 h dark cycle at 20 ± 2 °C. Total RNA was extracted from the seedlings of the above plants using the MiniBEST Plant RNA Extraction Kit (TaKaRa, Dalian, China). First-strand cDNAs were synthesized with 2 µg total RNAs using a PrimeScript^TM^ II first-strand cDNA Synthesis Kit (TaKaRa) according to the manufacturer’s protocol. A 420 bp cDNA fragment corresponding to bases 994–1414 of the *GhCLA1* gene (KJ123647) was amplified and the resulting PCR product was cloned into *Kpn* Ι-cut pTRV2 to obtain the recombinant plasmid pTRV2-*GhCLA1*. At the same time, a 459 bp cDNA fragment corresponding to bases 281–740 of the *GhSOS1* gene (NM-001327028) was amplified and the resulting PCR product was cloned into *Kpn* Ι-cut pTRV2 to obtain the recombinant plasmid pTRV2-*GhSOS1*. The full-length coding region of *TrSOS1* was amplified from cDNA as described in our previous study [29], and the resulting PCR product was ligated into the plasmid *Sma* Ι-cut pSuper^+^1300 to obtain the recombinant plasmid pSuper^+^1300-*TrSOS1*. Subsequently, the open reading frames of *TrSOS1*, *GhSOS1*, and *AtSOS1* (AT2G01980) were amplified from cDNA. Then, the resulting PCR products were ligated into the *EcoR* Ι-cut pYES2 plasmid to obtain the recombinant plasmids pYES2-*TrSOS1*, pYES2-*GhSOS1*, and pYES2-*AtSOS1*. The primers used for the abovementioned gene amplification are listed in Table 1. After sequence verification, and by the freeze–thaw method [36], the recombinant plasmid pTRV2-*GhCLA1* or pTRV2-*GhSOS1* was transformed into *A. tumefaciens* GV3101, and the recombinant plasmid pSuper^+^1300-*TrSOS1* was transformed into *A. rhizogenes* K599. The recombinant plasmids pYES2-*TrSOS1*, pYES2-*GhSOS1*, and pYES2-*AtSOS1* were respectively transformed into the NHA1-deleted yeast mutant strain *ena1nha1* using the PEG/LiAc procedure and transformants were identified by PCR-based methods [37].

### 4.3. Construction of GhSOS1-VIGS Cotton Plants

*GhSOS1*-VIGS cotton plants were obtained according to the methods described by Gao et al. [38] using a 1:1 mixture of *A. tumefaciens* GV3101 containing pTRV1 and pTRV2-*GhSOS1* which were pelleted and resuspended in infiltration culture containing 10 mM MgCl_2_, 10 mM MES, and 200 µM acetosyringone, and two fully expanded cotyledons of one-week-old plants were infiltrated using a needleless syringe. In the same way, the cotton plants infected with *A. tumefaciens* GV3101 containing pTRV1 and pTRV2 for 10 days were used as the empty-vector control, and the plants infected with *A. tumefaciens* GV3101 containing pTRV1 and pTRV2-*GhCLA1* for 10 days were used as a parallel control (*GhCLA1*-VIGS), which showed the photo bleaching phenotype.

### 4.4. Construction of hrTrSOS1-OE/GhSOS1-VIGS Cotton Plants

Cotton plants were first infected with *A. rhizogenes* strain K599 containing pSuper^+^1300-*TrSOS1* according to the methods of Wei et al. [31], using the transformation with the empty vector pSuper^+^1300 as a negative control. Total DNA isolated from the roots and leaves of the empty-vector- or *TrSOS1*-transformed plants was identified by the PCR-based method of An et al. [39]. Then, the empty-vector- or *TrSOS1*-transformed cotton plants were infected once with *A. tumefaciens* GV3101 containing pTRV1 and pTRV2-*GhSOS1* for 10 days as indicated before to obtain *hrGhSOS1*-VIGS or *hrTrSOS1*-OE/*GhSOS1*-VIGS cotton plants.

### 4.5. Semi-Quantitative RT-PCR Analysis

For semi-quantitative RT-PCR analysis of *GhCLA1* or *GhSOS1* gene silencing, total RNA was isolated from the leaves of *GhCLA1*-VIGS cotton plants, the roots and leaves of *GhSOS1*-VIGS, *hrGhSOS1*-VIGS or *hrTrSOS1*-OE/*GhSOS1*-VIGS cotton plants, respectively, then the first-strand cDNAs were synthesized as indicated previously. The housekeeping gene *GhActin9* was used as an internal control. The amplification program for this work was performed at 94 °C for 3 min, followed by 30 cycles of 94 °C for 30 s, 55 °C for 30 s, 72 °C for 40 s, and a final extension of 72 °C for 10 min. The primers used for *GhCLA1*, *GhSOS1*, and *GhActin9* are listed in Table 1.

### 4.6. Salt-Tolerance Tests of GhSOS1-VIGS or TrSOS1-OE/GhSOS1-VIGS Cotton Plants

The above-obtained vector and *GhSOS1*-VIGS, *hrGhSOS1*-VIGS and *hrTrSOS1*-OE/*GhSOS1*-VIGS cotton plants were treated with 1/2 Hoagland solution containing 0 or 200 mM NaCl for seven days. After being photographed, the plants were fully rinsed in distilled water, fixed at 105 °C for 10 min, and dried to a constant weight at 80 °C to measure the dry weight per plant. Leaf chlorophyll content (SPAD value) was measured with a chlorophyll meter (SPAD-502PLUS, Konica Minolta Holdings, Inc., Tokyo, Japan), and leaf relative water content (RWC) was determined using the following formula: RWC = (FW − DW)/(TW − DW) × 100%, and relative electrolytic leakage (REL) in roots and leaves was calculated as: (C_1_ − C_w_)/(C_2_ − C_w_) × 100%, both as described by Tian et al. [40]. The malondialdehyde (MDA) contents in the leaves and roots were measured according to the method described by Jouve et al. [41], and the amount of MDA was calculated from the following formula: C = 6.45(A_532_ − A_600_) − 0.56A_450_. Na^+^ and K^+^ contents in the roots, stems, and leaves of cotton plants were measured according to the method of Chen et al. [42], using a flame spectrophotometer (AP1200 type, Shanghai, China).

### 4.7. Complementation Test of TrSOS1, GhSOS1, and AtSOS1 in Yeast Mutant

According to our previous methods [19] with minor modifications, ten-fold serial dilutions (starting at OD_550_ ≈ 0.5) of each yeast culture, including G19 (*ena1*) as a positive control, and the mutants ANT3 (*ena1nha1*), ANT3-*TrSOS1* (*ena1nha1+ TrSOS1*), ANT3-*GhSOS1* (*ena1nha1+GhSOS1*), and ANT3-*AtSOS1* (*ena1nha1+ AtSOS1*), were plated on YPD (1% yeast extract, 2% peptone, and 2% dextrose) medium, YPG (1% yeast extract, 2% peptone, and 2% galactose) medium, or arginine–phosphate medium (AP medium: 10 mM L-arginine, 8 mM H_3_PO_4_, 2 mM MgSO_4_, 0.2 mM CaCl_2_, 2% glucose, vitamins, and trace elements, pH 6.5) plus NaCl (130 mM) [43] and KCl (1 mM), and the plates were incubated at 30 °C for three days and then photographed. For Na^+^ and K^+^ content measurements, the yeast cells were grown in liquid AP medium plus NaCl (130 mM) and KCl (1 mM) and collected during the exponential growth phase (OD_550_ ≈ 0.2). The Na^+^ and K^+^ contents in the yeast cells were determined using a flame spectrophotometer as described above.

### 4.8. Statistical Analyses

Data were expressed as the mean ± SD for each treatment (*n* = 3, except for the measurement of dry weight per plant, where *n* = 6, semi-quantitative RT-PCR for *GhSOS1* expression in the roots and leaves of cotton plants, where *n* = 5) using SPSS software (ver. 20.0., International Business Machines Corporation, Armonk, NY, USA), and differences among means were determined by Duncan’s test at *p* ≤ 0.05.

## 5. Conclusions

The recretohalophyte *Tamarix TrSOS1* gene conferred enhanced salt tolerance to transgenic hairy root composite cotton seedlings exhibiting *GhSOS1*-VIGS by maintaining enhanced root vigor and leaf RWC and a reduced Na^+^ content and Na^+^/K^+^ ratio in the roots, stems, and leaves. A yeast mutant complementation test elucidated the contribution of *SOS1* genes from the recretohalophyte *Tamarix* salt-tolerant glycophyte cotton, and salt-sensitive glycophyte *Arabidopsis* to salt tolerance: *TrSOS1* ≥ *GhSOS1* > *AtSOS1*. Therefore, the contribution of the recretohalophyte *Tamarix TrSOS1* gene to salt tolerance is very similar to that of other plants’ *SOS1* gene: The direct contribution is the participation in the enhancement of Na^+^ extrusion in plants (mainly through roots, and possibly through leaves) under salt stress and the regulation of Na^+^ and K^+^ homeostasis, while the indirect contribution is the improvement of root vigor, reduction in the cell membrane damage in the roots and leaves, and thus the promotion of plant growth. Moreover, this study effectively indicates that the salt-tolerant function of the *SOS1* gene of a certain kind of plant may be related to its habitat or salt-tolerant habit. This finding suggests that the *SOS1* gene of halophytes or salt-tolerant crops should be given preference for utilization in the germplasm innovation and genetic improvement of salt-tolerant plants or crops by genetic engineering in the future.

## Figures and Tables

**Figure 1 ijms-20-02930-f001:**
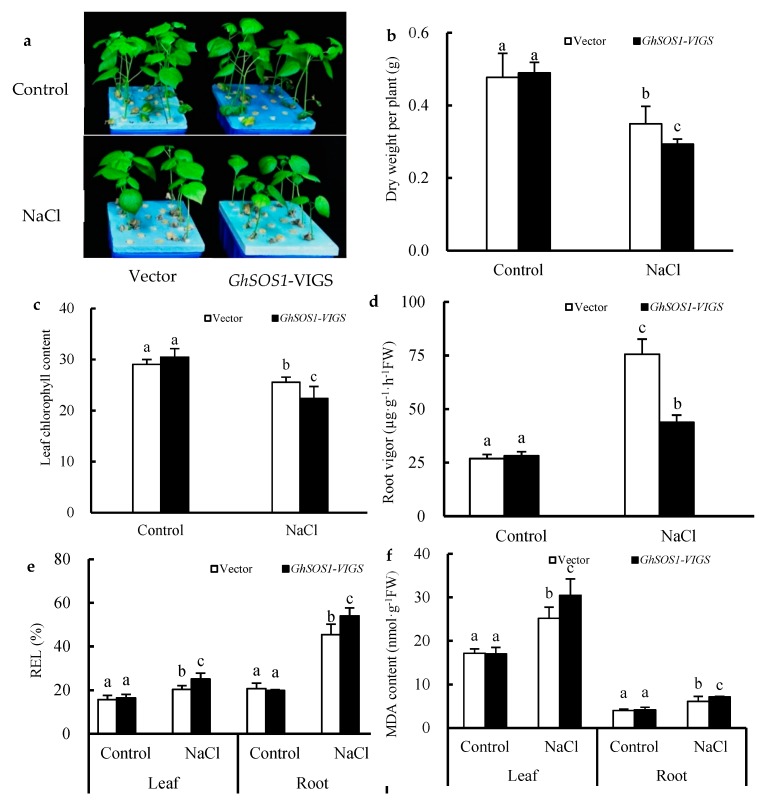
Effects of salt treatment on the growth (**a**), dry weight per plant (**b**), leaf chlorophyll content (**c**), root vigor (**d**), relative electrolytic leakage (REL) value (**e**), and MDA content (**f**) in the leaves and roots of *GhSOS1*-VIGS (virus-induced gene silencing) cotton plants. Different lowercases in the same group indicate the significant differences (*p* ≤ 0.05).

**Figure 2 ijms-20-02930-f002:**
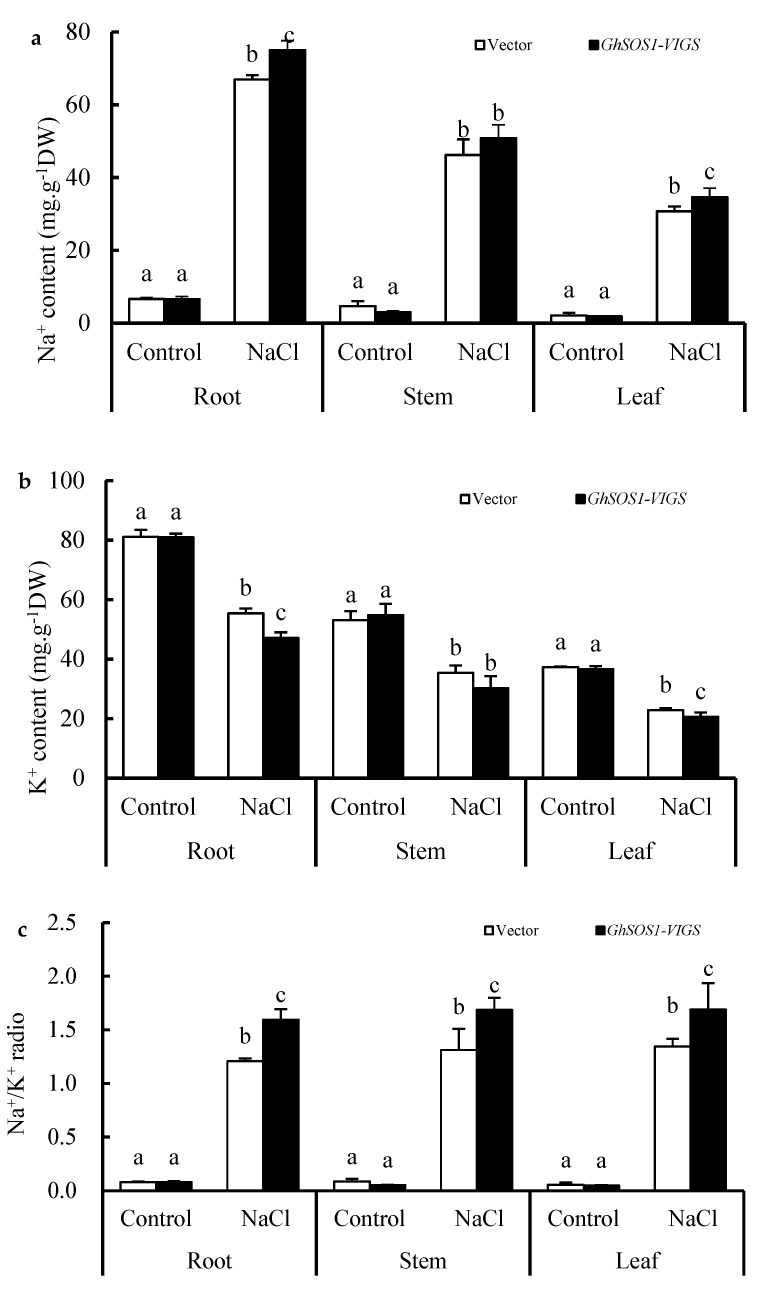
Changes in contents of Na^+^ (**a**) and K^+^ (**b**), and Na^+^/K^+^ ratio (**c**) in the roots, stems, and leaves of the *GhSOS1*-VIGS cotton plants under salt stress. Different lowercases in the same group indicate the significant differences (*p* ≤ 0.05).

**Figure 3 ijms-20-02930-f003:**
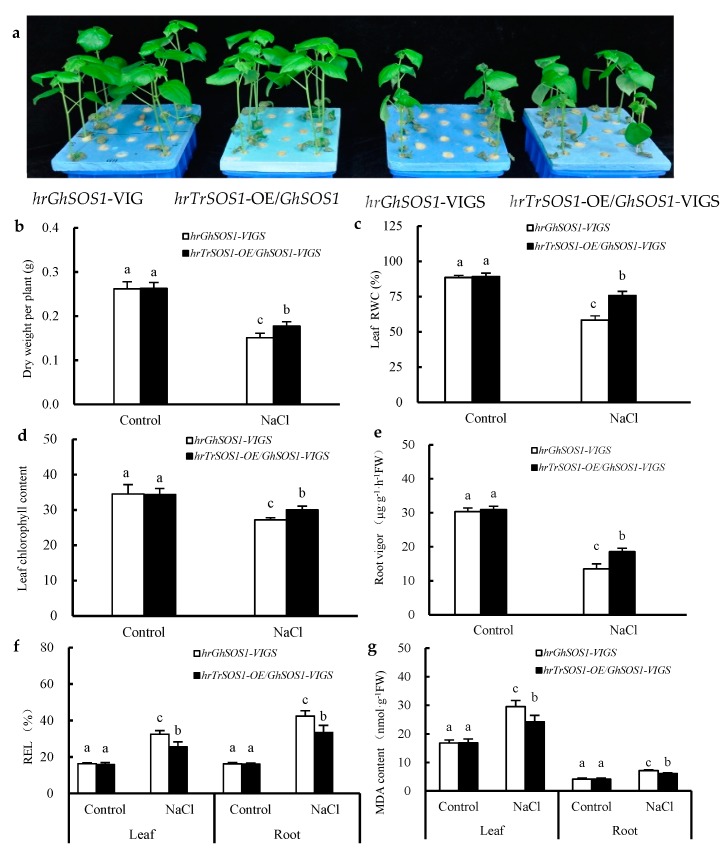
Effects of salt treatment on growth phenotype (**a**), dry weight per plant (**b**), leaf relative water content (RWC) (**c**), leaf chlorophyll content (**d**), root vigor (**e**), REL (**f**), and MDA content (**g**) in the leaves and roots of hairy root composite cotton plants of *hrGhSOS1*-VIGS or *hrTrSOS1*-OE/*GhSOS1*-VIGS. Different lowercases in the same group indicate the significant differences (*p* ≤ 0.05).

**Figure 4 ijms-20-02930-f004:**
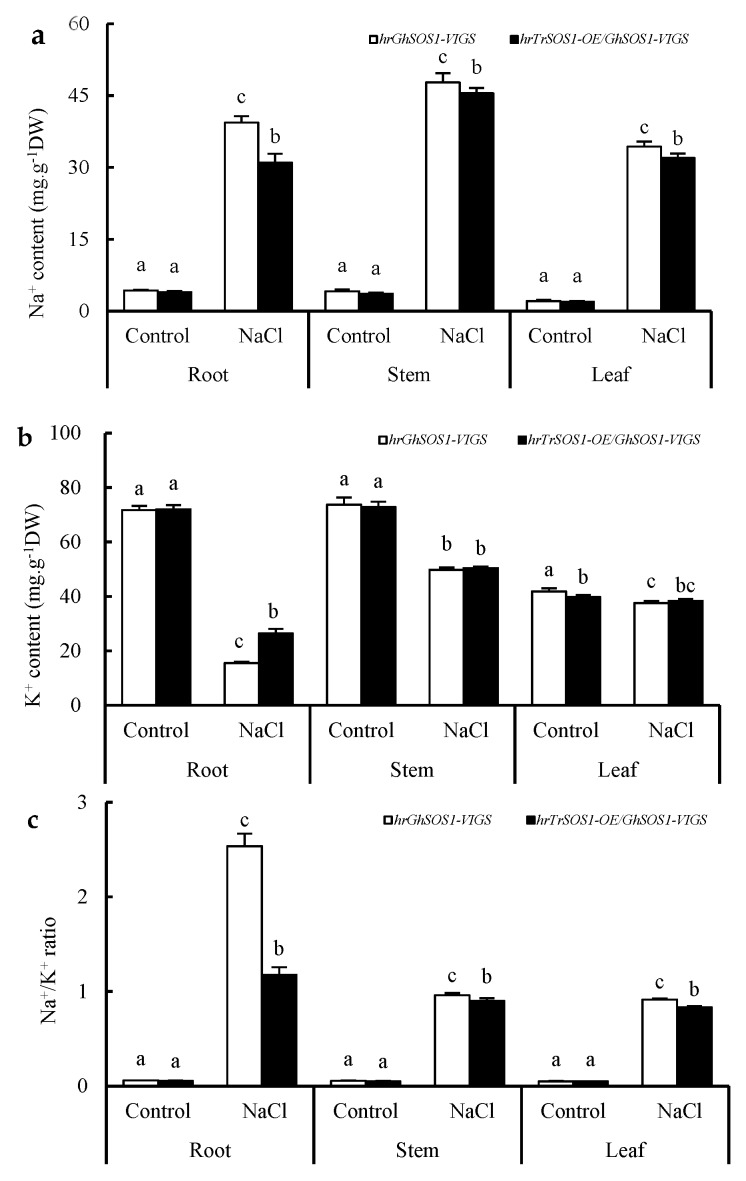
Effect of salt stress on contents of Na^+^ (**a**), K^+^ (**b**), and Na^+^/K^+^ ratio (**c**) in the roots, stems, and leaves of hairy root composite cotton plants of *hrGhSOS1*-VIGS or *hrTrSOS1*-OE/*GhSOS1*-VIGS. Different lowercases in the same group indicate the significant differences (*p* ≤ 0.05).

**Figure 5 ijms-20-02930-f005:**
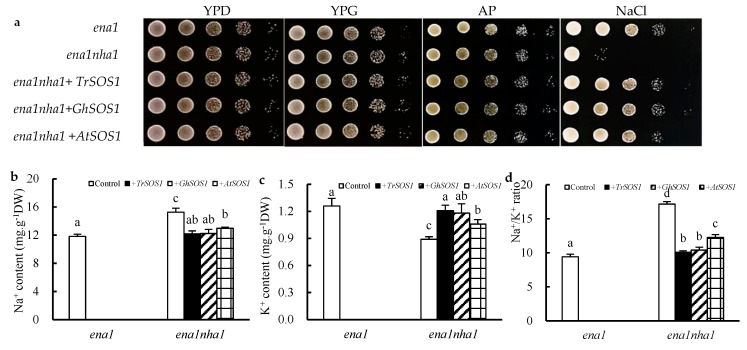
Effects on the growth phenotype (**a**), contents of Na^+^ (**b**), K^+^ (**c**), and Na^+^/K^+^ ratio (**d**) of the yeast strains *ena1*, *ena1nha1*, and *ena1nha1* complemented by *TrSOS1, GhSOS1, and AtSOS1* under NaCl treatment. Different lowercases in the same group indicate the significant differences (*p* ≤ 0.05).

**Table 1 ijms-20-02930-t001:** Primers for gene analysis.

Purposes	Genes		Primer Sequences (5′ to 3′)
Plant VIGS	*GhCLA1*	F	CTCCATGGGGATCCGGTACCCACAACATCGATGATTTAG
		R	AGACGCGTGAGCTCGGTACCATGATGAGTAGATTGCAC
	*GhSOS1*	F	CTCCATGGGGATCCGGTACCTATTAGCTGTTTTTCTACCCGC
		R	AGACGCGTGAGCTCGGTACCATAAATCCAAGCCACAAAACAG
Plant expression	*TrSOS1*	F	GCTTCTGCAGGGGCCCGGGATGGCAGCGGTGTCCGAATT
		R	TTTAAATGTCGACCCCGGGTCAAGAAGCATGACGGAAAGATAGC
Semi-quantitative RT-PCR	*GhCLA1*	F	TCTTACCCTCACAAAATCTTGAC
		R	GCATGGATGGCAACAATATT
	*GhSOS1*	F	ATCCAGGCAGCATACTGGGA
		R	ATGAAGTTGCCGTCGTGCTA
	*Ghactin9*	F	TGAAATATCCCATTGAGCACG
		R	TGTAGTTTCATGGATTCCAGCAG
Yeast complementation	*TrSOS1*	F	CCGCCAGTGTGCTGGAATTCATGGCAGCGGTGTCCGAATT
		R	GATGGATATCTGCAGAATTCTCAAGAAGCATGACGGAAAGATAGC
	*GhSOS1*	F	CCGCCAGTGTGCTGGAATTCATGGAGGAAGTGAAAGAGTATC
		R	GATGGATATCTGCAGAATTCTTAAGAAGCCTGGTGGAAT
	*AtSOS1*	F	CCGCCAGTGTGCTGGAATTCATGACGACTGTAATCGACGC
		R	GATGGATATCTGCAGAATTCTCATAGATCGTTCCTGAAAACG

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
