# Peer review of "The Recretohalophyte Tamarix TrSOS1 Gene Confers Enhanced Salt Tolerance to Transgenic Hairy Root Composite Cotton Seedlings Exhibiting Virus-Induced Gene Silencing of GhSOS1"

_ijms, 2019, doi:10.3390/ijms20122930_

Round 1
Reviewer 1 Report
In this MS author explain about the recretohalophyte Tamarix TrSOS1 gene involvement in salt tolerance to transgenic hairy root composite cotton seedlings exhibiting virus-induced gene silencing of GhSOS1. The overall manuscript is very well written. However, for the betterment of the MS, I have suggestions. Major. 1. The introduction part is short please add some references of study with proven sal stress tolerance. Like a. Ectopic expression of OsSta2 enhances salt stress tolerance in rice. b. Overexpression of a MYB Family Gene, OsMYB6, Increases Drought and Salinity Stress Tolerance in Transgenic Rice. 2. Also, add proline content, some target gene expression. Minor L107 markedly to marked
Author Response
Response to Reviewer 1 Comments
In this MS author explain about the recretohalophyte Tamarix TrSOS1 gene involvement in salt tolerance to transgenic hairy root composite cotton seedlings exhibiting virus-induced gene silencing of GhSOS1. The overall manuscript is very well written. However, for the betterment of the MS, I have suggestions.
Point 1: The introduction part is short please add some references of study with proven salt stress tolerance. Like a. Ectopic expression of OsSta2 enhances salt stress tolerance in rice.b. Overexpression of a MYB Family Gene, OsMYB6, Increases Drought and Salinity Stress Tolerance in Transgenic Rice.
Response 1: Thank you very much for the suggestion. We have added the reference a. [Kumar et al.(2017) Ectopic Expression of OsSta2 Enhances Salt Stress Tolerance in Rice. Front. Plant Sci. 8:316. doi: 10.3389/fpls.2017.00316] in the Introduction part, and the relevant text statement was also improved.
With regard to the ref. b [Tang et al. (2019) Overexpression of a MYB Family Gene, OsMYB6, Increases Drought and Salinity Stress Tolerance in Transgenic Rice. Front. Plant Sci.,10:168. doi: 10.3389/fpls.2019.00168], we have checked and read it, because MYB is a large TF family in plants, there are over 183 members in rice [Chen et al. (2006) The MYB transcription factor superfamily of Arabidopsis: expression analysis and phylogenetic comparison with the rice MYB family. Plant Mol Biol 60: 107–124.] Over the past decade, it has been reported that the most of MYB genes are involve in the response of plants to diverse abiotic stress. They can play an important role in the drought and salinity stress tolerance of rice. But our manuscript mainly focused on the comparison of contribution of SOS1 genes from the recretohalophyte Tamarix, salt tolerant glycophyte cotton, and salt sensitive glycophyte Arabidopsis to salt tolerance. The relationship between the abovementioned ref. b and our study seems not very direct. In addition, the number of existing references in our manuscript is more than 40, so we hope not add the references on MYB family for the time being. We expect to be forgiven if anything goes wrong.
Point 2: Add proline content, some target gene expression. Minor L107 markedly to marked.
Response 2: Thanks, this is a reasonable advice. Proline acts as one of the mostly common osmolytes in plants under adverse environments, and some related target genes for its biosynthesis, such as P5CS1 (coding pyrroline-5-carboxylate synthetase) and P5CDH (coding pyrroline-5-carboxylate dehydrogenase), should be paid due attention. We feel very sorry that, for reasons of the limited time and heavy workload, we mainly focused on the growth phenotype and related physiological parameters, such as plant dry weight, leaf RWC and chlorophyll content, root vigor, REL value and MDA content in the roots and leaves, and Na+, K+, and Na+/K+ ratio in the roots, stems and leaves of GhSOS1-VIGS or TrSOS1-OE/GhSOS1-VIGS cotton seedlings, and didn't conduct the proline content and some target gene expression in the present manuscript. We will seriously considerate this advice in our future research. In addition, L107 of “markedly” was changed to “marked”.
Reviewer 2 Report
Soil salinity is a major abiotic stress that decreases plant growth and productivity. Recently, it was reported that plants overexpressing SOS1 or SOS 3 have significantly increased salt tolerancein Arabidopsis, in sweetpotato or in tall fescue .
Tamarix, a family of woody recretohalophyte species, is widely distributed in the saline soils of drought-stricken areas of Central Asia and China.
The main aim of the present MS is to provide a novel theoretical basis for the exploitation of the superior salt tolerance function of halophyte SOS1 gene and its future prior utilization in germplasm innovation and genetic improvement of crops such as cotton by genetic engineering.
The MS has novelty, the expresiment is well-designed and the presented results are correct.
The whole MS is nicely written, the conlusion is compact and coherent.
These data markedly contribute to the knowledge about salt tolerance in plants.
Minor comments, see also other recent studies, in order to complete the introduction:
https://www.frontiersin.org/articles/10.3389/fpls.2019.00299/full
https://www.nature.com/articles/srep20582
http://horizon.documentation.ird.fr/exl-doc/pleins_textes/divers16-12/010068741.pdf
Author Response
Response to Reviewer 2 Comments
Soil salinity is a major abiotic stress that decreases plant growth and productivity. Recently, it was reported that plants overexpressing SOS1 or SOS3 have significantly increased salt tolerance in Arabidopsis, in sweet potato or in tall fescue.
Tamarix, a family of woody recretohalophyte species, is widely distributed in the saline soils of drought-stricken areas of Central Asia and China.
The main aim of the present MS is to provide a novel theoretical basis for the exploitation of the superior salt tolerance function of halophyte SOS1 gene and its future prior utilization in germplasm innovation and genetic improvement of crops such as cotton by genetic engineering.
The MS has novelty, the experiment is well-designed and the presented results are correct.
The whole MS is nicely written, the conclusion is compact and coherent.
These data markedly contribute to the knowledge about salt tolerance in plants.
Point: Minor comments, see also other recent studies, in order to complete the introduction:
https://www.frontiersin.org/articles/10.3389/fpls.2019.00299/full
https://www.nature.com/articles/srep20582
http://horizon.documentation.ird.fr/exl-doc/pleins_textes/divers16-12/010068741.pdf
Response 1: Thank you very much for your suggestion. We have downloaded and read these references, and added the ref. Zhang et al. (2016) Genetic regulation of salt stress tolerance revealed by RNA-Seq in cotton diploid wild species, Gossypium davidsonii. Sci. Rep. 6:20582, and Hanin et al. (2016) New Insights on Plant Salt Tolerance Mechanisms and Their Potential Use for Breeding. Front. Plant Sci. 7:1787. doi: 10.3389/fpls.2016.01787 in the Introduction section, and the relevant text statement was also improved. We look forward to your understanding and support.
Round 2
Reviewer 1 Report
I am Happy with the author's reply. MS can be accepted in its current format.